# Uncertainty-Aware Test-Time Augmented Ensemble of BERTs for Classification of Common Mental Illnesses on Social Media Posts

**Pratinav Seth** *
seth.pratinav@gmail.com

**Mihir Agarwal** *
mihiragarwal1423@gmail.com

## Abstract

Given the current state of the world, because of existing situations around the world, millions of people suffering from mental illnesses feel isolated and unable to receive help in person. Psychological studies have shown that our state of mind can manifest itself in the linguistic features we use to communicate. People have increasingly turned to online platforms to express themselves and seek help with their conditions. Deep learning methods have been commonly used to identify and analyze mental health conditions from various sources of information, including social media. Still, they face challenges, including a lack of reliability and overconfidence in predictions resulting in the poor calibration of the models. To solve these issues, We propose UATTA-EB: Uncertainty-Aware Test-Time Augmented Ensembling of BERTs for producing reliable and well-calibrated predictions to classify six possible types of mental illnesses. Code is available at https://github.com/ptnv-s/UATTA-EB.

## 1  Introduction & Previous Works

Mental illness is a type of health condition that alters a person's intellect, feelings, or behavior (or all three) and has been shown to affect an individual's physical health. Depression, schizophrenia, attention deficit hyperactivity disorder (ADHD), autism spectrum disorder (ASD), and other mental health issues are widespread today, with an estimated 450 million worldwide suffering from such problems(Sayers (2001)). Unlike other chronic conditions, which are diagnosed using research facility tests and measurements, mental illnesses are regularly diagnosed using an individual's self-report surveys designed to detect specific patterns of feelings or social interactions(Hamilton (1967)). People have recently started communicating and seeking assistance with mental health difficulties on social media. Several methods for studying mental health have been developed through monitoring user behavior on social media. Initially, many researchers concentrated on data from Facebook, and Twitter (Benton et al. (2017)), Instagram, Flickr, and other social media platforms to research mental health.

Various approaches to mental health text analysis have been used, from traditional ML to advanced DP, including (Abusaa et al. (2004),Amjad et al. (2021),Cohan et al. (2018) and Ameer et al. (2022)) with many deep learning-based approaches have been introduced recently. While Data Augmentation is frequently performed during model training, recent work has shown that using Data Augmentation at inference, or Test-Time Augmentation (TTA)(Shanmugam et al. (2021)), can improve Model Accuracy, Robustness, and Uncertainty Estimates. In classification tasks, a poorly calibrated network can place a high probability on one of the classes, even when the predicted class is incorrect (Ayhan et al. (2020)). The uncertainty estimation issue is especially important to trust confident model predictions for screening automation and referring uncertain cases for manual intervention of a medical expert.

Even though fine-tuned pre-trained transformers (Devlin et al. (2019)) have achieved state-of-the-art accuracy on text classification tasks, they still suffer from the same over-confidence problem. Bayesian probability theory offers a sound mathematical framework to design machine learning

---

*Department of Data Science & Computer Applications, Manipal Institute of Technology, Manipal Academy of Higher Education, Manipal, India

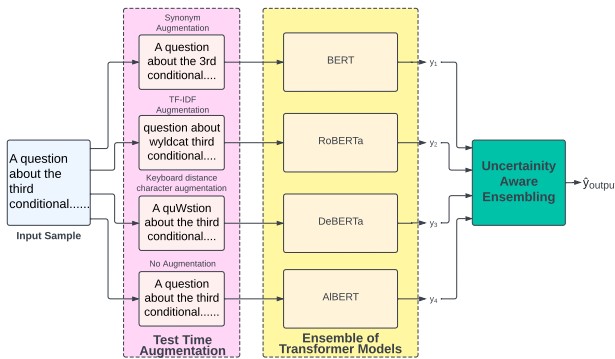

Figure 1: UATTA-EB: Model Architecture

models with an inherent and explicit notion of uncertainty (Yang & Fevens (2021)) with Multiple probabilistic and Bayesian methods, (Graves (2011)), and Non-Bayesian such as (Lakshminarayanan et al. (2017)) have been proposed to quantify the uncertainty estimates. However, Bayesian methods aren't model-agnostic methods and require a specific architecture. To solve this problem, we introduce a reliable end-to-end architecture UATTA-EB for Classifying Common Mental Illnesses on Social Media Posts to produce reliable and well-calibrated predictions.

## 2 METHODOLOGY

Given a training set $(x_n, y_n)_{n=1}^N$ consisting of N i.i.d examples, we define an end-to-end model architecture as depicted in Figure 1. The architecture consists of multiple stages. The text input is augmented in the first stage using the Test Time Augmentation(Lu et al. (2022)), which is used as input for the models, which produce individual predictions, which are then ensembled using Uncertainty-Aware Ensembling by weighting the predictions based on their uncertainty to get the final output $\hat{y}$.

In Test Time Augmentation, the train and test samples are randomly augmented, which accounts for the noise that may arise in real-world data, thus making the model more robust. The transformations used in our approach using (Ma (2019)) include Synonym augmentation for $30\%$ of the sample words, TF-IDF word augmentation for $5\%$ of the sample words, and keyboard augmentation of individual characters at $5\%$ of the sample.

We calculate the uncertainty estimation of the output of each model to create an ensemble architecture that is aware of the uncertainty of the predictions while ensembling. The output prediction $\hat{y}_i$ where $i \in (1, N)$ is weighted with its uncertainty. For the final prediction, $\hat{y}$, quantifies an uncertainty weighted average (Jaskari et al. (2022)).

The uncertainty weights are formulated by taking an inverse of the independent uncertainty metric. Hence, $\hat{y}(\mathbf{x}_n)$ outputs the final prediction corresponding to $n^{th}$ data point. Here the uncertainty associated with each prediction is quantified using a modified version of LLFU(Lakara et al. (2021)). For further details on methodology kindly refer to Appendix A.

## 3 EXPERIMENTS & RESULTS

We used (Murarka et al. (2021)), a mental illnesses dataset extracted from the Reddit social media platform which consists of 16703 social media posts and includes the title, the post, and the illness. The task is to classify into one of the six mental illnesses - Bipolar, ADHD, anxiety, depression, PTSD, and None using both titles and posts.

We have used various models for our experiments and conducted extensive experimentation for shortlisting the models based on Accuracy and F1-Score. The shortlisted variants are BERT, RoBERTa, DeBERTa, and AlBERT. We evaluate the results from the baseline models and the proposed architecture over a fixed test set using Accuracy and F1 Score.

Table 1: Evaluation Metrics and Uncertainty Metrics for various model architectures on the test set.

| Model Architecture | Accuracy | F1 Score | ECE | MCE | Brier Score |
|---|---|---|---|---|---|
| Baseline (RoBERTa) | **0.86** | **0.86** | 0.039 | 0.398 | 0.25 |
| UA-ENS(Uncertainty Aware Ensemble) | 0.85 | **0.86** | 0.023 | 0.189 | **0.23** |
| UATTA-EB(Proposed) | 0.85 | **0.86** | **0.018** | **0.122** | **0.23** |

We use three metrics to quantify the uncertainty - Expected Calibration Error (ECE), Maximum Calibration Error (MCE), and Brier Score. Table 1 shows the results we observe all architectures have quite similar accuracy and F1 score. However, it is observed from the uncertainty metrics that the baseline models are not well-calibrated and reliable enough. However, using our proposed architecture, we observe a significant drop in Calibration Metrics without a trade-off in performance, thus producing quite reliable and less overconfident predictions. For further details on the experiments and results kindly refer to Appendix A

## 4 URM STATEMENT

The authors acknowledge that at least one key author of this work meets the URM criteria of the ICLR 2023 Tiny Papers Track. Both authors' Age lies outside the range of 30-50 years; geographically are not located in North America, Western Europe, the UK, or East Asia, and they are of non-White Race. The second author is a First-time submitter to a conference of the stature of ICLR.

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

## A APPENDIX

### A.1 METHODOLOGY

Given a training set $(x_n, y_n)_{n=1}^{N}$ consisting of N i.i.d examples, we define an end-to-end model architecture as depicted in Figure 1. The architecture consists of multiple stages. The text input is augmented in the first stage using the Test Time Augmentation, which is used as input for the models for the Transformer models (Wolf et al. (2020)), which produce individual predictions, which are then ensembled using Uncertainty-Aware Ensembling by weighting the predictions based on their uncertainty to get the final output $\hat{y}$.

#### A.1.1 TEST TIME AUGMENTATION

Test-Time Augmentation (Lu et al. (2022)Wang et al. (2019)) is the aggregation of predictions across transformed examples of test inputs and is an established technique to improve performance and produce well-calibrated classification models (Lu et al. (2022)). It can deliver consistent improvements over current state-of-the-art approaches.

The train and test samples are randomly augmented, which accounts for the noise that may arise in real-world data, thus making the model more robust. The transformations used in our approach using (Ma (2019)) include Synonym augmentation for $30\%$ of the sample words, TF-IDF word augmentation for $5\%$ of the sample words, and keyboard augmentation of individual characters at $5\%$ of the sample.

#### A.1.2 UNCERTAINITY AWARE ENSEMBLE

We calculate the uncertainty estimation of the output of each model to create an ensemble architecture that is aware of the uncertainty of the predictions while ensembling. The output prediction $\hat{y}_i$ where $i \in (1, N)$ is weighted with its uncertainty. For the final prediction, $\hat{y}$, the weighted ensemble quantifies an uncertainty weighted average (Jaskari et al. (2022)).

$$\hat{y}(\mathbf{x}_n) = \frac{\sum_{j=1}^{k} \frac{1}{\sigma_{h^j}(\mathbf{x}_n)} \hat{y}_{h^j}(\mathbf{x}_n)}{\sum_{j=1}^{k} \frac{1}{\sigma_{h^j}(\mathbf{x}_n)}} \tag{1}$$

Table 2: Training, Validation and Test set Accuracy for various BERT variants.

| Model | Train accuracy | Validation Accuracy | Test Accuracy |
|---|---|---|---|
| BERT | 0.92 | 0.85 | 0.84 |
| ROBERTA | 0.91 | 0.86 | 0.86 |
| DEBERTA | 0.84 | 0.83 | 0.82 |
| ALBERT | 0.86 | 0.84 | 0.83 |
| DISTILBERT | 0.18 | 0.17 | 0.17 |
| ELECTRA | 0.18 | 0.17 | 0.17 |

Table 3: Evaluation Metrics and Uncertainty Metrics for various model architectures on the test set.

| Model Architecture | Accuracy | F1 Score | ECE | MCE | Brier Score |
|---|---|---|---|---|---|
| Baseline (BERT) | 0.84 | 0.84 | 0.046 | 0.259 | 0.29 |
| Baseline (RoBERTa) | **0.86** | **0.86** | 0.039 | 0.398 | 0.25 |
| Baseline (DeBERTa) | 0.82 | 0.82 | 0.050 | 0.332 | 0.34 |
| Baseline (ALBERT) | 0.83 | 0.83 | 0.047 | 0.272 | 0.31 |
| UA-ENS(Uncertainty Aware Ensemble) | 0.85 | **0.86** | 0.023 | 0.189 | **0.23** |
| UATTA-EB(Proposed) | 0.85 | **0.86** | **0.018** | **0.122** | **0.23** |

Here, $x_n$ denotes an nth input sample, $\hat{y}_{h^j}(x_n)$ is the output of nth input sample for the $j_{th}$ model prediction, $\sigma_{h^j}$ is the estimated uncertainty corresponding to predictions from the $j_{th}$ model prediction.

The uncertainty weights are formulated by taking an inverse of the independent uncertainty metric. (Sarawgi et al. (2021)). Hence, $\hat{y}(\mathbf{x}_n)$ outputs the final prediction corresponding to $n^{th}$ data point. Here the uncertainty associated with each prediction is quantified using a modified version of LLFU(Lakara et al. (2021)).

$$\sigma_{h^j} = max(0, \frac{1}{2}log(2\pi\sigma^2(x_n))) + \frac{(y_j(x_n) - \mu(x_n))^2}{2\sigma^2(x_n)} \qquad (2)$$

where $y_j(x_n)$ - denotes the prediction corresponding to $jth$ model, $\mu(x_n)$ refers to the mode of predictions from all the models in ensemble and $\sigma^2(x_n)$ refers to standard deviation of predictions of models in ensemble for the $n^{th}$ data point.

## A.2 EXPERIMENTS

We conduct a series of experiments to study whether the proposed architecture can correctly classify mental illnesses. We have used four pretrained Language model backbones each followed by a fully connected layer and Softmax from HuggingFace Wolf et al. (2020) in the Ensemble along with their respective tokenizers. The maximum sequence length is 512. We used CrossEntropy loss with the Adam Kingma & Ba (2015) optimizer with a $10^{-5}$ learning rate and batch size of 16.

### A.2.1 DATASET

We used Murarka et al. (2021), a mental illnesses dataset extracted from the Reddit social media platform. The task is to classify the post into one of the six following mental illnesses Bipolar, ADHD, anxiety, depression, PTSD, and None.

The dataset consists of 16703 social media posts from Reddit, which include the title, the post, and the illness. The dataset is split into three parts, training - 13727, validation - 1488, and test - 1488 posts. We used both titles and posts combined for our analysis.

### A.2.2 SYSTEM DESCRIPTION AND ENSEMBLING

We have used various models for our experiments, including: BERT Devlin et al. (2019) is a transformer-based Vaswani et al. (2017) language model. RoBERTa Liu et al. (2019) a faster variation of BERT. DeBERTa He et al. (2021) variant of RoBERTa. AlBERT Lan et al. (2020) is a lightweight Self-supervised variant of BERT. We performed ablations with many of the present well-known language models — including Electra Clark et al. (2020), and DistilBERT Sanh et al. (2019).

We conducted extensive experimentation and observed some models to perform substantially better than others. We shortlisted the models based on Accuracy and F1-Score on the evaluation set for our proposed architecture. The shortlisted variants are BERT, RoBERTa, DeBERTa, and AlBERT.

Model Selection On the training set, we fine-tuned pre-trained BERT, RoBERTa, DeBERTa, AlBERT, Electra, and DistilBERT from Huggingface Wolf et al. (2020) individually.

We observed that DistilBert and Electra performed worse than others, as shown in Table3, comparing accuracy in training, validation, and test sets. Hence, we shortlisted Bert, Roberta, Deberta, and Albert as the ensemble architecture's backbone.

## A.3 EVALUATION AND UNCERTAINTY METRICS

The model performance is evaluated using Accuracy and F1 Score.We use three metrics to quantify the uncertainty - Expected Calibration Error (ECE) (Nixon et al. (2019)), Maximum Calibration Error (MCE) (Widmann et al. (2019)), and Brier Score (Rufibach (2010)).

### A.3.1 EXPECTED CALIBRATION ERROR

The Expected Calibration Error (ECE)(Nixon et al. (2019)) is a weighted average over the absolute confidence difference of the predictions of a model. It is defined as

$$ECE = \sum_{i \in \mathcal{B}_{\updownarrow}} \frac{|B_m|}{n} |acc(B_m) - conf(B_m)| \tag{3}$$

where

$$acc(B_m) = \frac{1}{|B_m|} \sum_{i \in B_m} 1(y_i = y_t) \tag{4}$$

$$conf(B_m) = \frac{1}{|B_m|} \sum_{i \in B_m} p_i \tag{5}$$

where $conf(B_m)$ is just the average confidence/probability of predictions in that bin, and $acc(B_m)$ is the fraction of the correctly classified examples Bm.

### A.3.2 MAXIMUM CALIBRATION ERROR

The Maximum Calibration Error (MCE) (Widmann et al. (2019)) focuses more on high-risk applications where the maximum confidence difference is more important than the average. It is then defined as:

$$MCE = max_m |acc(B_m) - conf(B_m)| \tag{6}$$

### A.3.3 BRIER SCORE

The Brier Score (Rufibach (2010)) is a strictly proper score function or scoring rule that measures the accuracy of probabilistic predictions.

$$BrierScore = \frac{1}{n} \sum_{t=1}^{n} (f_t - o_t)^2 \tag{7}$$

where ft is the probability that was forecast, ot is the actual outcome of the event at instance t (0 if it does not happen and one if it does happen), and N is the number of forecasting instances.

## A.4 RESULTS

We evaluate the results from the baseline models and the proposed architecture over a fixed test set using Accuracy and F1 Score. We use three metrics to quantify the uncertainty - Expected Calibration Error (ECE)(Nixon et al. (2019)), Maximum Calibration Error (MCE)(Widmann et al. (2019)), and Brier Score(Rufibach (2010)).

Table 3 shows that all the baseline models and proposed model architecture have quite similar accuracy and F1 score. However, it is observed from the uncertainty metrics that the baseline models are not well-calibrated and reliable enough. However, using our proposed architecture, we observe an instant drop in Uncertainty Metrics, thus producing quite reliable and well-calibrated models.

## A.5 Social Impact

The strategy described here tries to address the problem of overconfident model predictions for the early detection of multiple mental illnesses on social media posts. Detection of mental illness using social media posts has shown much growth and promise in recent years with the application of large language models. However, for rigorous evaluation, its applicability in high-stakes healthcare decisions must include systematic uncertainty quantification and calibration.

This framework also substantially helps in risk assessment, reducing the probability of probable misdiagnosis by calibrating more broad models. Our research is intended to improve the current method for the early detection of mental illnesses and to provide reliable predictions.

