# OpenReview forum: "Uncertainty-Aware Test-Time Augmented Ensemble of BERTs for Classification of Common Mental Illnesses on Social Media Posts"
_ICLR.cc/2023/TinyPapers — Submitted to Tiny Papers @ ICLR 2023_

### Official Review · Reviewer_dAGX · 2023-03-31

**Confidence:** 3

**Summary Of Contributions:**

The paper proposed an ensemble-based method to classify mental illnesses. The proposed method achieves similar performance with baseline models (measured in accuracy and F1) but has better uncertainty (measured in ECE, MCE and Brier).

**Rating:**

Great Start (GS): a submission which meets some of the reviewing criteria but has room for improvement

**Strengths And Weaknesses:**

Strengths:
1. The paper explains the motivation, what existing methods lack, and their proposed method well.
2. The problem itself seems to be of interest for people working in mental health industry.
3. The result, despite tested only on one dataset (Murarka et al. 2021), seems promising.

Weaknesses:
1. The contribution of this work might be very limited to this specific application (or even this dataset). I am not convinced that the proposed method would be of interest to broader audiences outside of mental health.
2. Experiment could include other datasets as well, such as the ones used in Benton et al. (2017).

**Suggested Changes:**

1. Experiment with more datasets for mental illnesses.
2. Although the target application is to classify mental health, such high uncertainty issue is not only applied to this specific fields. Does the proposed method can benefit other applications as well?

---

### Meta-Review · Area_Chair_pDMG · 2023-04-04

**Recommendation:** Invite to archive
**Confidence:** 3

**Metareview:**

The paper explains the motivation, existing works, and the proposed architecture well. The method may be interesting in the specific domain and the results look promising. However, the framing and the scope are limited to a specific and narrow application only (mental health), which limits the applicability and potential impact of the paper.


**Summary:**

This paper combines Test Time Augmentation and Uncertainty-Aware Ensembling to improve uncertainty on the classification of common mental illnesses.

**Reason For Not Giving A Higher Recommendation:**

The scope is probably too narrow on the mental health application only and may have a limited impact among the audience.

**Reason For Not Giving A Lower Recommendation:**

The writing explains the work well and the results are promising the specific application considered.

---

### Decision · Program_Chairs · 2023-04-08

Invite to archive

---

> ### Author Response · Authors · 2023-06-01
> **Thanks a lot**
>
> We want to thank the reviewers for their feedback. We will try to incorporate the suggestions made by them.
> We would like to opt-in for archival.
> Thanks a lot!